# Air Quality Impact Assessment of a Waste-to-Energy Plant: Modelling Results vs. Monitored Data

**Giovanni Lonati [1],\*** , **Stefano Cernuschi [1]** and **Paolo Giani [2]**

1   Department of Civil and Environmental Engineering, Politecnico di Milano, 20133 Milano, Italy; stefano.cernuschi@polimi.it
2   Department of Civil & Environmental Engineering & Earth Sciences, University of Notre Dame, South Bend, IN 46556, USA; pgiani@nd.edu
\*   Correspondence: giovanni.lonati@polimi.it

**Abstract:** The impact of the emissions from a municipal Waste-to-Energy (WtE) plant in Northern Italy on local air quality was assessed using the CALMET-CALPUFF atmospheric dispersion modelling system. Model simulations were based on hourly emission rates measured by continuous stack monitoring systems and considered both air quality-regulated pollutants (nitrogen oxides, particulate matter, toxic elements, benzo(a)pyrene), and other trace pollutants typical of WtE plants (dioxins, furans, and mercury). The model results were compared to both long-term observations from the air quality monitoring network and with short-term measurements from dedicated monitoring campaigns in the vicinity of the WtE plant, in both warm and cold season conditions. Modelling and observational results showed that the estimated plant contributions are very limited. This suggests that the observed concentration levels were the result of the contribution of all the sources distributed over the area and that they were not solely driven by the activity of the plant. Estimated contributions from the plant's emissions were usually at least two orders of magnitudes lower than the ambient levels at the nearest monitoring site and even lower at the farthest sites.

**Keywords:** air quality; waste to energy; trace pollutants; measured emissions; monitored data; atmospheric dispersion modelling

## 1. Introduction

Energy recovery via the incineration of waste residual from separate collection is part of the waste management and circular economy strategies of the European Union [1–3]. While reducing the mass and volume of waste to be disposed of, Waste-to-Energy (WtE) plants produce power or generate heat for district heating; modern WtE plants are typically combined heat and power (CHP) plants and trigeneration systems (i.e., the generation of electricity, heat and cold through the integration of heat pump systems) are under development.

Due to their atmospheric emissions, WtE plants frequently face strong opposition from local communities, making the siting of new plants an ongoing concern [4–7]. Interestingly, despite the progressively more stringent limits on atmospheric emissions (EU directive 2010/75) [8] and of the technological improvements in flue gas treatment [9,10], there is still considerable public concern regarding the potential adverse health effects of waste incineration, especially among the communities living near WtE plants. Indeed, associations between exposure to the emissions of waste incinerators and health outcomes (i.e., increased risk of lung/throat cancer or ischemic heart disease, non-Hodgkin's lymphoma, soft-tissue sarcoma) have been reported [11,12], but the findings were inconsistent because of poor methods of exposure characterization [13]. Recent review studies on the potential health effects of exposure to waste-related combustion emissions concluded that the evidence was insufficient to support the association between waste incineration processes and adverse health effects [14–18]. Thus, properly designed, operated, and controlled WtE

plants appear as a reasonable option for waste management and energy security with limited health impacts or risks [14].

Air quality monitoring and robust modelling, based on both accurate emissions assessment and proper atmospheric dispersion calculations, are required for a comprehensive evaluation of the real environmental impact of WtE plants and to correctly assess the related health risks correctly. Accurate model results can help in (i) estimating the contribution of the plant's emission to the current levels of air pollution and (ii) comparing the role of the WtE source with all the other sources affecting local air quality (i.e., traffic, domestic heating or biomass burning), which are often mistakenly considered as less harmful to human health by public perception [19]. Indeed, health risk assessment studies for new plants are usually based on the maximum mass flow rate of pollutants and provide upper-bound estimates of the impact of the WtE plants on air quality. Due to this cautious approach, these studies indicate an acceptable incremental risk for the exposed population but do not assess the real contribution of the plant's emissions to air pollution.

Because studies presenting a combined modelling–monitoring approach to assess the environmental impact of WtE plants are rare in the literature, this work intends to contribute insights by discussing the case of the WtE plant operated by Alto Vicentino Ambiente SpA in the city of Schio (Northern Italy, Veneto region) as a case study. Specifically, this study uses a combined modeling–monitoring approach to address the following goals:

To assess the real impact of the plant's emissions on local air quality based on actual emission data, in order to make the findings public knowledge for the benefit of the local population;

To estimate the contribution of the plant's emissions to the ambient levels of air quality-regulated pollutants routinely at the regional monitoring network site in the vicinity of the plant;

To compare model results with the ambient levels of non-regulated pollutants (i.e., mercury, dioxin and furans) measured during dedicated air quality monitoring campaigns at sites with different exposure to the plant's emissions.

In Section 2, the study area, the modelling system and its input emission and meteorological data are described, along with the air quality data available for comparison with modelling results. In Section 3, model results are discussed and compared with air quality data for regulated (i.e., nitrogen oxides, particulate matter, toxic elements, benzo(a)pyrene) and non-regulated pollutants (i.e., dioxins and furans, and mercury). Finally, Section 4 summarizes the main findings of this work, indicating its limitations and outlining future lines of research.

## 2. Materials and Methods

### 2.1. Study Area

The WtE plant is located in the industrial area of the municipality of Schio in the Veneto region, Northern Italy. The city of Schio is in a plain area at about 200 m a.s.l. (above sea level), open to the river Po plain to the South East but surrounded on all the other sides by the mountainous amphitheater of the Piccole Dolomiti mountains, with peaks as high as 1300 m a.s.l.

Model simulations were performed over a $20 \times 20$ km$^2$ computational domain centered on the plant location with a grid resolution of 125 m. In the UTM32-WGS84 system, the x-coordinate (West–East) ranges between 677 km and 697 km and the y-coordinate (South–North) ranges between 5055 km and 5075 km. For model simulations, additional discrete receptor points were located at the air quality monitoring station of Schio (R1: 684.28 km, 5064.95 km) and at sites where dedicated air quality monitoring campaigns have been performed (R2: 687.23 km, 5064.42 km; R3: 688.08 km, 5065.48 km; R4: 691.51 km, 5070.97 km). The modelling domain, including the plant location and the discrete receptors, is shown in Figure 1.

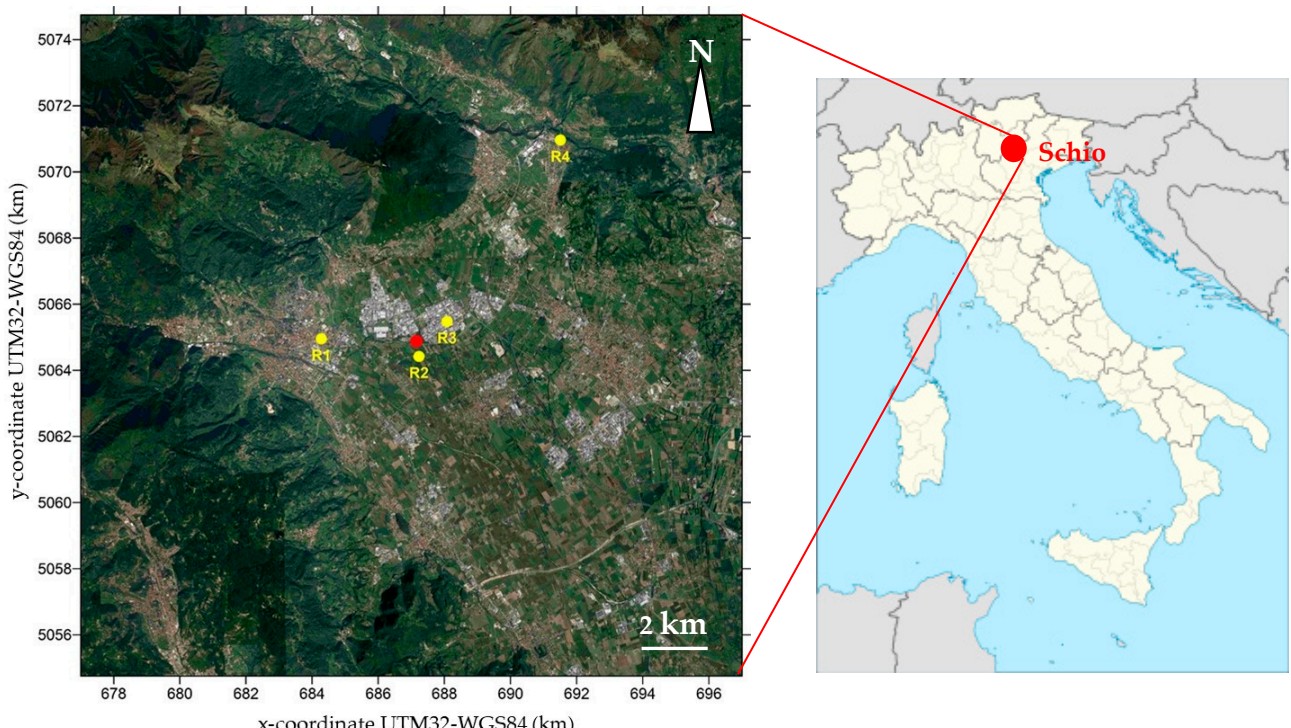

**Figure 1.** Modelling domain with plant (red dot) and discrete receptors' (yellow dots) location.

### 2.2. Modelling System

Atmospheric dispersion simulations were conducted using the CALMET/CALPUFF modelling system. CALMET [20] is a diagnostic three-dimensional model that reconstructs wind and temperature fields starting with meteorological measurements, orography and land use data. CALPUFF [21] is a non-steady-state Gaussian puff model that simulates the effects of time- and space-varying meteorological conditions on pollutant transport, transformation, and removal. The CALPUFF model was selected because it is particularly suitable for near-field assessment in complex dispersion conditions, including complex terrain (i.e., hilly or mountainous terrain, where geographically induced wind circulation effects may occur) and/or complex meteorological conditions, including stagnation/inversion conditions and light winds and calm conditions. These latter conditions frequently occur in Northern Italy's Po Valley plain [22,23]. CALPUFF output (i.e., the time series of 1 h concentrations at each grid node of the computational domain) was then processed by the CALPOST post-processor in order to obtain summary statistical data (i.e., average annual concentration, hourly maximum/daily average/percentile values) for graphical representation with maps showing the iso-concentration contour lines and for comparison with monitoring data and air quality limits.

#### 2.2.1. Emission Data

The plant has three separate combustion lines, all equipped with moving grate technology. Each line is followed by a flue gas treatment line composed of an electrostatic precipitator, a dry sorption reactor, a fabric filtration unit, and a catalytic Denox. Sodium bicarbonate and activated carbon are injected in the dry sorption reactor for acid gases and trace organic and inorganic pollutants control. After the treatment the flue gases from Line 1 and 2 are mixed and are released to the atmosphere from a 40 m tall stack (2 m diameter); another 40 m stack (1.3 m diameter) serves the third line (Table 1). Daily waste throughputs of the three lines are 72 tons day$^{-1}$, 60 tons day$^{-1}$, and 100 tons day$^{-1}$, respectively. The combustion lines are fed with residual residential waste from separate collections and with a small fraction of hospital waste (less than 5%). Continuous emission monitoring systems (CEMs) are installed on each line, which collect flue gas data (volumetric flow

rate, temperature, water vapor and oxygen content) and concentration data for the main pollutants, namely for nitrogen oxides (NOx) and particulate matter (PM), with an hourly resolution. Procedures set by EN14181:2014 European standard [24], which established quality assurance levels for automated CEM systems, are adopted to ensure data quality. Flue gas sampling campaigns are periodically performed (3 times a year) to monitor the emission of organic and inorganic toxic pollutants including As, Cd, Hg, Ni, Pb, speciated polycyclic aromatic hydrocarbons (PAHs), such as benzo(a)pyrene, and speciated dioxins and furans (PCDD/F). Sampling and analytical procedures follow the current ISO and EN standards.

**Table 1.** Stack locations and geometric features (UTM32-WGS84 coordinates, height, diameter), and mean and min–max range (square brackets) for the atmospheric release conditions (hourly flow rate, temperature, outlet speed).

| Parameter | Stack Line 1 and 2 | Stack Line 3 |
|---|---|---|
| Stack location x, y coordinate (km) | 687.152, 5064.883 | 687.165, 5064.868 |
| Stack height (m) | 40 | 40 |
| Stack diameter (m) | 1.6 | 1.3 |
| Actual flow rate ($m^3 \, h^{-1}$) | 89,730 [39,560–105,450] | 78,560 [48,415–99,930] |
| Temperature ($°C$) | 163.2 [136.1–176.2] | 141.1 [99.4–187.9] |
| Stack outlet speed ($m \, s^{-1}$) | 12.4 [5.5–14.6] | 16.4 [4.15–20.9] |

For model simulations, stack-tip features of the flue gas (temperature and exit velocity) were derived with hourly resolution by CEMs data collected in the reference period 1 May 2018–30 April 2019. This period was selected because it is the most recent period for which reliable local meteorological data were available. During the reference period, the combustion lines operated 24/7 with a fairly constant waste-feeding rate, except for shutdown and start-up phases within ordinary maintenance program periods (two times/year, two weeks each). However, at least two lines were always concurrently in operation.

Hourly variable mass flow rates of NOx and PM were directly determined from the CEMs data. Hourly variable mass flow rates of the toxic pollutants were determined from the product of the hourly variable volumetric flow rate from CEMs data and the average concentration measured during the periodic emission monitoring campaigns. Because only three campaigns were performed in the reference period, data from the two other campaigns conducted in 2019 were also used in order to enlarge the datasets and to obtain a more robust estimate of the stack concentrations. Additionally, as the three lines burn the same kind of waste (i.e., residual waste from separate collection) and have the same technologies, concentration data were pooled together and their distributions were analyzed in order to find a common representative average value.

For As and Cd, all data were below the minimum detection limit of 0.6 μg $m^{-3}$ (0 °C, 101.3 kPa, dry basis, 11% $O_2$); therefore, the representative concentration value was set at 0.6 μg $m^{-3}$, without any further investigation. For Hg, Ni, and Pb only a fraction of the data (78% for Hg, 54% for Ni, 50% for Pb) was below the detection limit and the data distributions could be analyzed; similarly, benzo(a)pyrene and dioxins and furans (PCDD/F) concentration data distributions were analyzed. For these pollutants, concentration data were described reasonably well by lognormal distributions (Figure S1 in Supplementary Materials). The estimated parameters of the lognormal distributions (i.e., geometric mean and standard deviation) are reported in Table 2. The arithmetic means computed for the lognormal distributions, also reported in Table 2, were adopted as representative stack concentration values.

**Table 2.** Parameters of the fitted lognormal distributions for Hg, Ni, Pb, benzo(a)pyrene (BaP), and PCDD/F concentrations (Geometric and arithmetic mean: $\mu g \, m^{-3}$ for Hg, Ni, Pb; $ng \, m^{-3}$ for BaP; $ng_{I-TEQ} \, m^{-3}$ for PCDD/F; all concentrations referred at 0 °C, 101.3 kPa, dry basis, 11% $O_2$. Geometric standard deviation: dimensionless).

| Parameter | Hg | Ni | Pb | BaP | PCDD/F |
|---|---|---|---|---|---|
| Geometric mean | 0.09 | 0.58 | 0.73 | 1.93 | 0.0027 |
| Geom. St. Dev. | 10.8 | 3.16 | 3.35 | 1.52 | 2.69 |
| Arithmetic mean | 0.28 | 1.12 | 1.52 | 2.38 | 0.0044 |

Flue gas sampling campaigns also provided data on the size distribution of PM. Consistent with the dust emission control technology of the plant, PM emission were entirely made up by PM10 (particle diameter < 10 μm). However, no conclusive information on smaller size cuts could be drawn. Thus, PM emissions were simulated as PM10, assuming a lognormal distribution for the particle size with 0.48 μm as the mode and 2 μm as the standard deviation, according to the CALPUFF model default values.

2.2.2. Meteorological Data

The meteorological data required by the CALPUFF model were provided by the Environmental Agency of Veneto region (ARPAV) for the reference period. Measurements from all the stations available within the simulation domain were used (five monitoring sites), with the closest site to the plant at about 7 km; in addition, upper air data from vertical soundings of the atmosphere at the stations located in the Po plain (Milano-Linate, Udine-Rivolto, Bologna-San Pietro Capofiume) were incorporated. The meteorological fields were generated with the CALMET preprocessor on a $20 \times 20 \, km^2$ domain centered on the plant. The descriptive parameters of atmospheric stability, the height of the mixed layer and three-dimensional fields of atmospheric temperature and wind speed and wind direction were calculated for ten vertical layers up to an altitude of 3000 m with 1 h temporal resolution and 1 km spatial resolution.

The ground-level (10 m) annual wind rose computed for the plant location (Figure 2) showed relatively weak winds (average speed of about 0.9 m s$^{-1}$), mainly blowing from the North West quadrant (about 50% of the time) and in particular from the North West (about 16%), North North West (about 26%) and North (about 9%) sectors. Annual frequencies in the orders of 5–7% were observed for winds blowing from the south-eastern sectors, with a slight prevalence for the South-East sector. For all other directions, annual frequencies were of less than 3%, with almost no winds from the Western and South-Western quadrants. The wind roses at higher altitudes (Figure S2) substantially maintained the same features in the first four layers (up to 200 m from the ground level), but with gradual intensification of the wind speed (average speed 1.8 m s$^{-1}$ a 160 m). At higher altitudes, the increase in the frequency of winds from North was first observed (200–300 m), and then the progressive clockwise rotation of the wind rose beginning in the 300–1000 m layer, characterized by the predominance of winds coming from the sectors around East.

The ground-level wind regime did not show any relevant seasonality (Figure S3), but clearly displayed a local circulation due to the surrounding horography. During nighttime, north-westerly winds blew towards the plain, whereas easterly and south-easterly winds blew towards the mountains during daytime (Figure S4). Overall, such wind conditions were in reasonable agreement with those observed for locations in the Po Valley at the base of the mountains, where the local wind is usually characterized by light breeze circulation.

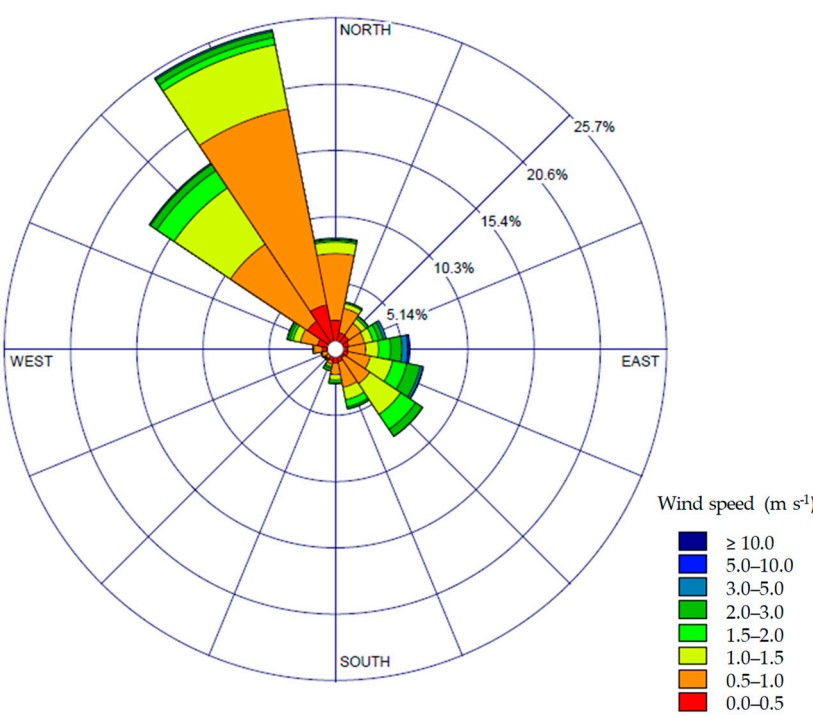

**Figure 2.** Ground-level (10 m) wind rose at the plant location for the simulation reference period (1 May 2018–30 April 2019).

Almost half of the reference period (45%, Figure 2) was characterized by stable conditions (classes E and F of the Pasquill–Gifford categories), whereas weak (class C) and moderate instability conditions (Class B) were present 22.5% and 17.3% of the time, respectively. Strong instability conditions (Class A) were found only 3.7% of the time. The remaining hours were characterized by a near-neutral atmosphere (11.6%, Class D).

A joint analysis of the average daily trend of wind speed and direction, atmospheric stability, and mixing layer height highlighted the following features (Figure 3):

- Wind speed was substantially uniform throughout the day, usually below 1 m s$^{-1}$ but with a slight increase in the afternoon;
- Winds were usually aligned in the NNW-SSE direction, blowing towards the plain during nighttime, early morning, and early evening hours and towards the mountain during daytime hours;
- Conditions of neutral and stable atmosphere occurred from the evening to the early hours of the morning, followed by a progressive increase in instability until mid-afternoon;
- The evolution of the height of the mixed layer followed a temporal profile anticorrelated with atmospheric stability, with shallow values in the orders of 50 m in the evening and night hours, gradually increasing in the morning up to maximum values of the orders of 1000–1200 m in the hours before sunset.

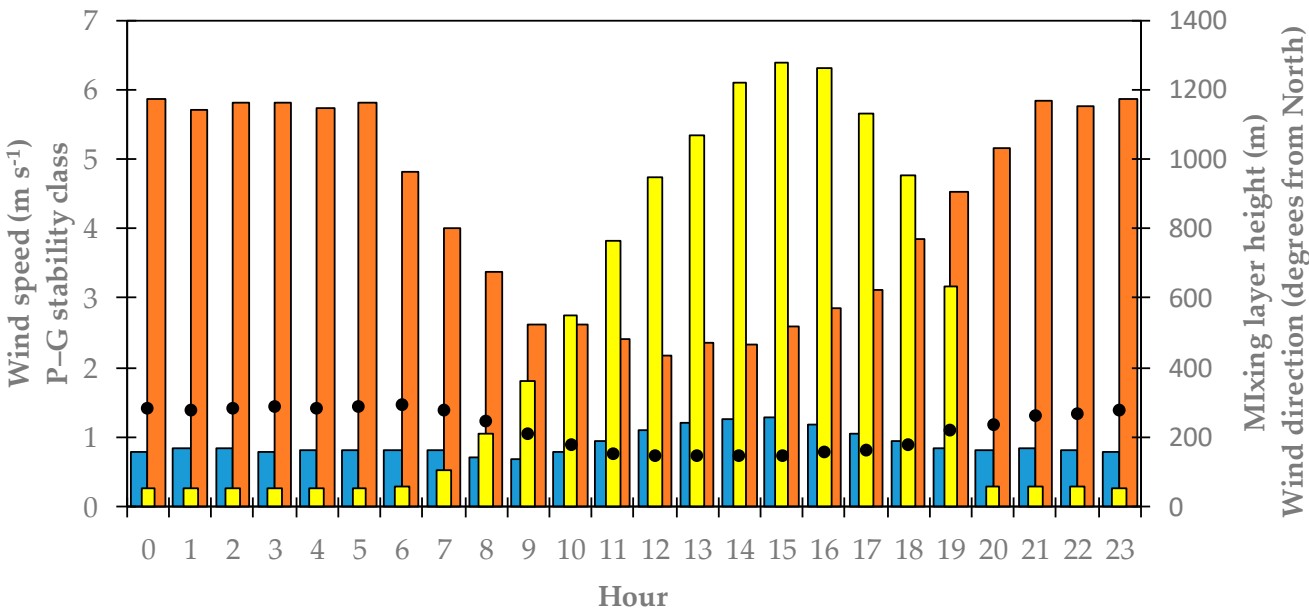

**Figure 3.** Average daily patterns of wind speed (blue bars, left axis), atmospheric stability (orange bars, left axis, P–G classes: class A = 1, class F = 6), wind direction (black dots, right axis), mixing layer height (yellow bars, right axis) for the simulation reference period.

### 2.3. Air Quality Data

The results of the model simulations were compared with air quality data available in the study area for the reference period. These data mainly derived from routine monitoring activity by ARPAV at the air quality monitoring station of Schio. Additionally, further data were available from short-period dedicated campaigns performed both by ARPAV [25] and by the plant managing company (AVA) at three monitoring sites around the plant. At the monitoring station (Site R1), $NO_2$ and PM10 were continuously monitored with an hourly ($NO_2$) and daily (PM10) resolution; and PM10 samples were separately collected for an analytical determination of the daily average concentration of the regulatory elements (As, Cd, Ni, Pb) and benzo(a)pyrene. For the reference period, 70 and 134 daily data points were available for the four toxic elements and for benzo(a)pyrene (BaP), respectively. Short-period campaigns were performed at site R2, both by ARPAV and AVA, and at site R3 and R4, by AVA only. Site R2 is suitable for identifying the impact of the plant, because it is the downwind sensitive receptor (permanently inhabited location) closest to the plant; site R3 is representative of the impact of the industrial area of Schio as a whole; site R4 represents a reasonable background situation, which is not significantly affected by the emissions of the plant, because of its location and distance from the plant (about 8 km direct distance to the North-East). Each monitoring campaign covered one week, according to the schedule summarized in Table 3, collecting PM samples for toxic pollutants detection (As, Cd, Ni, Pb, BaP, PCDD/F). Total gaseous mercury (TGM) measurements were performed at sites R2 and R4 during AVA campaigns. The PM sampling and analytical methods followed the regulatory protocols while ambient air measurements for TGM were performed with an automated system (Mercury Ultratracer UT-3000) that combines a gold trap amalgamation module with an atomic absorption spectroscopy detector for mercury vapour.

**Table 3.** Schedule of the short-period campaigns at the monitoring sites R2, R3, and R4.

| Site | ARPAV Campaign | AVA Campaign (Warm Season) | AVA Campaign (Cold Season) |
|---|---|---|---|
| R2 | 4 June 2018–11 June 2018 | 11 September 2018–18 September 2018<br>18 September 2018–26 September 2018 | 18 February 2019–25 February 2019<br>25 February 2019–4 March 2019 |
| R3 | - | 18 September 2018–26 September 2018 | 25 February 2019–4 March 2019 |
| R4 | - | 11 September 2018–18 September 2018 | 18 February 2019–25 February 2019 |

## 3. Results

The spatial distribution of ground-level concentrations due to the emissions of the WtE plant in the simulation domain is similar among all the pollutants, because it is driven by the meteorological conditions in the reference period. In particular, consistent with the local wind regime, the spatial distribution of the mean annual concentration values presented a shape essentially aligned along the North North West–South South East direction. The most significant influence of the plant emissions was within a $2 \times 2$ km$^2$ area around the plant and the maximum concentration estimated was at about 150 m North of the plant itself (Figure 4).

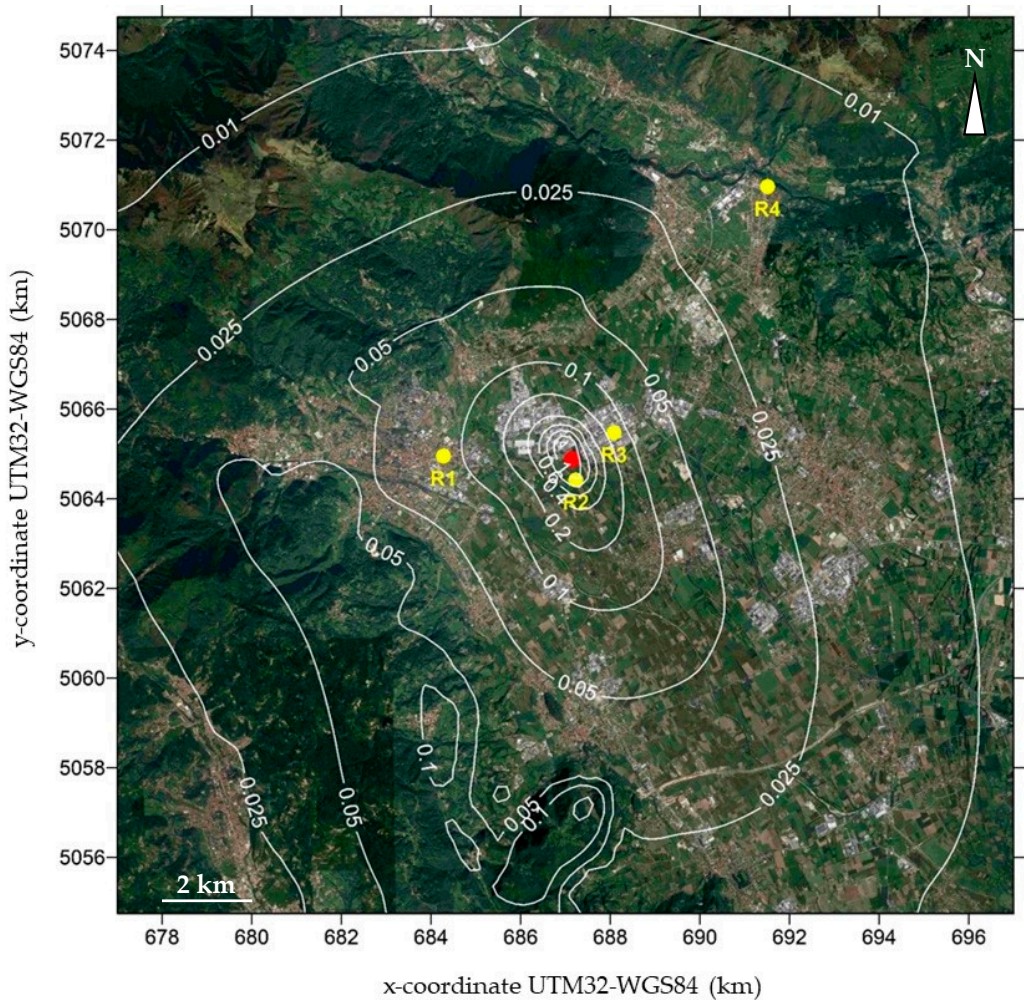

**Figure 4.** Spatial distribution of NO$_2$ annual mean concentration (µg m$^{-3}$) estimated as contribution of plant emissions to ambient levels.

The maximum mean annual values estimated as the contribution of the plant were at least three orders of magnitude lower than the corresponding air quality limits and up to

five orders for benzo(a)pyrene (Table 4). For $NO_2$, which was cautiously estimated assuming the complete oxidation of the NOx emitted by the plant, the gap between the maximum contribution (2.7 µg m$^{-3}$) and the annual limit value (40 µg m$^{-3}$) was less relevant, but was still of more than one order of magnitude (max/limit ratio = 6.75%). For PCDD/F, which is currently not regulated by air quality standards, the maximum contribution of the plant (1.5 × 10$^{-1}$ fg$_{I-TEQ}$ m$^{-3}$) was three orders of magnitude lower than the German guideline value of 150 fg$_{I-TEQ}$ m$^{-3}$, which also accounts for PCB-DL [26]. A large gap between the maximum estimated values and the short-term air quality limits was also observed. For $NO_2$, the maximum 1 h concentration in the simulation domain (5.5 µg m$^{-3}$) was almost 40 times lower than the regulatory value (200 µg m$^{-3}$ as 99.8 percentile of hourly concentrations); for $PM_{10}$, the maximum daily concentration (0.007 µg m$^{-3}$) was about four orders of magnitude lower than the limit (50 µg m$^{-3}$ as 90.4 percentile of daily concentrations).

**Table 4.** Annual mean concentrations: air quality limits (EU Directive 2008/50/EC) [27] and maximum estimated values in the modelling domain as WtE plant contribution.

| Pollutant | Air Quality Limit | Maximum Estimated |
|---|---|---|
| $NO_2$ (µg m$^{-3}$) | 40 | 2.7 |
| $PM_{10}$ (µg m$^{-3}$) | 40 | 2.0 × 10$^{-2}$ |
| BaP (ng m$^{-3}$) | 1 | 5.8 × 10$^{-5}$ |
| As (ng m$^{-3}$) | 6 | 9.2 × 10$^{-3}$ |
| Cd (ng m$^{-3}$) | 5 | 9.2 × 10$^{-3}$ |
| Ni (ng m$^{-3}$) | 20 | 3.7 × 10$^{-2}$ |
| Pb (µg m$^{-3}$) | 0.5 | 3.3 × 10$^{-5}$ |

Site R2 was most affected by the plant emissions, with annual mean values of about 40% of the estimated maximum concentrations; at the other monitoring sites, lower impacts were estimated, of 3–5% of the maximum for sites R1 and R3, and of 0.7% for site R4. In the following subsections, a comparison of the model results with air quality data is first presented for site R1, where long-term measurements were available, and then for the other three sites, where short-term campaigns were performed.

*3.1. Air Quality Monitoring Site R1*

The time pattern of the observed and modelled hourly $NO_2$ and daily PM10 concentrations are presented in Figure 5. The comparison highlighted the marginal contribution of the plant's emission to the ambient levels of both the pollutants, and in particular of PM10. The observed hourly $NO_2$ values were in the 2–100 µg m$^{-3}$ range, with an annual mean concentration of 20.8 µg m$^{-3}$; the model results were in the 0–6 µg m$^{-3}$ range with an estimated contribution of 0.08 µg m$^{-3}$ to the annual average concentration (Figure 5a,b). The hourly values estimated by the model were mostly lower than 1 µg m$^{-3}$, with a few episodes reaching values of the order of 3–4 µg m$^{-3}$ and a maximum of 5.6 µg m$^{-3}$. The ratio between the estimated contribution of the plant and the observed values was generally less than 1%. When site R1 was right downwind of the WtE plant due to easterly winds (during about 40 h), the ratio was greater than 10%, with a maximum contribution of 26%. The PM10 values were in the 2–128 µg m$^{-3}$ range, with an annual mean concentration of 24.8 µg m$^{-3}$. The model results were in the 0–7 × 10$^{-3}$ µg m$^{-3}$ range and estimated a 5 × 10$^{-4}$ µg m$^{-3}$ contribution to the annual average concentration (Figure 5c,d). The contribution of WtE plant emissions was practically negligible, as suggested by the 0.07% maximum value for the ratio between estimated and observed daily mean concentrations. Similar contributions to the annual mean of $NO_2$ (0.08 µg m$^{-3}$) and PM10 (5.2 × 10$^{-4}$ µg m$^{-3}$) were estimated by dispersion modelling for the emissions of a WtE plant with the same features (daily waste throughput, emission control technology, stack height) located near Milan [19].

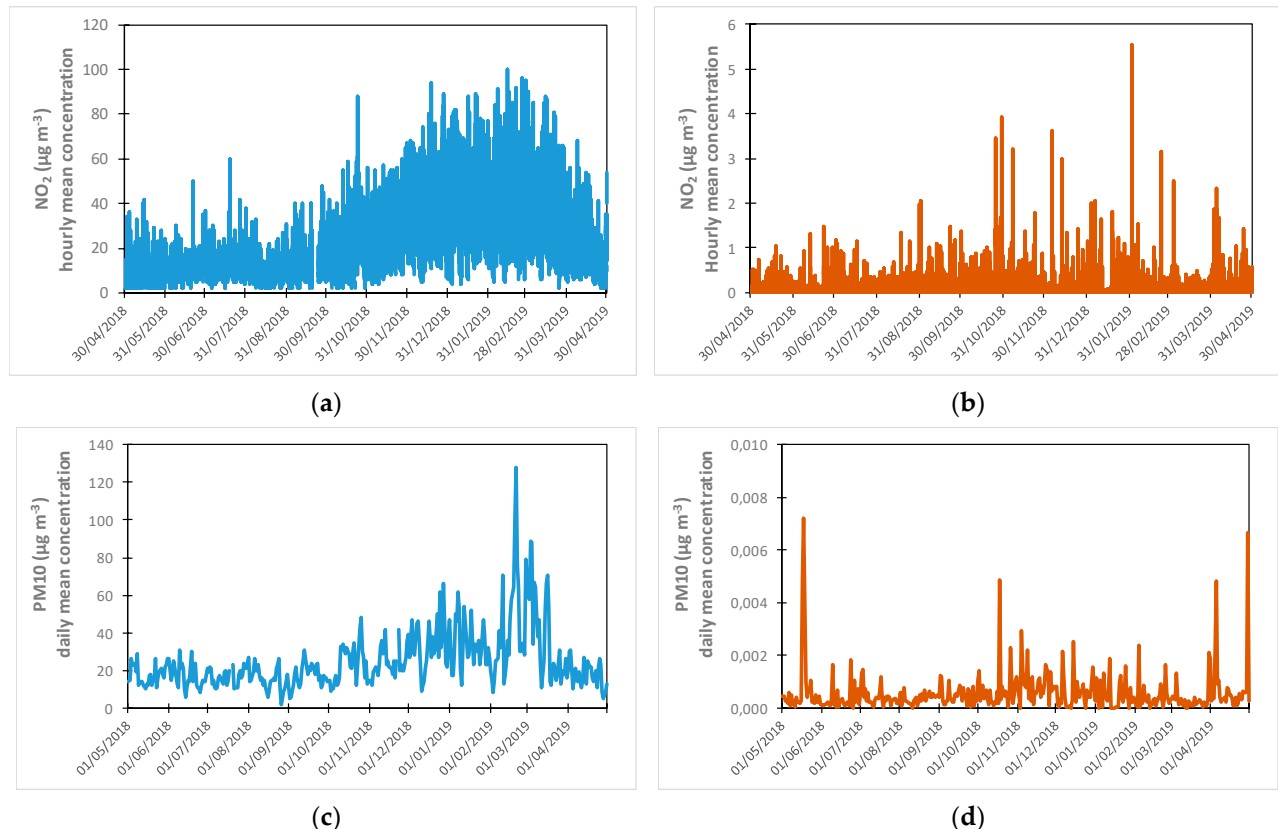

**Figure 5.** Time pattern of observed and modelled hourly $NO_2$ panel (**a**,**b**) and daily PM10 panel (**c**,**d**) concentrations for the reference period (1 May 2018–30 April 2019) at site R1.

The observed and modelled seasonal mean values for the reference period and the ranges for the daily average concentrations of the four toxic elements (As, Cd, Ni, Pb) and benzo(a)pyrene (BaP) are reported in Table 5; the same table also shows their mean and maximum relative contributions to the ambient levels due to the plant's emission.

**Table 5.** Mean and min-max range [square brackets] for the measured (Meas.) and modelled (Mod.) daily average concentrations (ng m$^{-3}$) and mean and maximum relative contributions of the plant's emission to the ambient levels of toxic elements and benzo(a)pyrene at site R1. (CS: cold season; WS warm season).

| Parameter | As | Cd | Ni | Pb | BaP |
|---|---|---|---|---|---|
| Meas. CS | <0.25 [<0.25] | 0.14 [0.05–0.6] | 2.2 [1.1–4.2] | 5.1 [1.4–13.6] | 0.31 [0.08–0.74] |
| Mod. CS | $5.2 \times 10^{-4}$ [$2.4 \times 10^{-5}$–$1.6 \times 10^{-3}$] | $5.2 \times 10^{-4}$ [$2.4 \times 10^{-5}$–$1.6 \times 10^{-3}$] | $9.6 \times 10^{-4}$ [$4.4 \times 10^{-5}$–$2.9 \times 10^{-3}$] | $1.3 \times 10^{-3}$ [$6.0 \times 10^{-5}$–$4.0 \times 10^{-3}$] | $2.3 \times 10^{-6}$ [$1.8 \times 10^{-7}$–$5.1 \times 10^{-6}$] |
| Meas. WS | <0.25 [<0.25] | 0.07 [0.05–0.3] | 1.4 [0.5–2.8] | 2.9 [1.4–6.8] | 0.02 [0.005–0.08] |
| Mod. WS | $4.0 \times 10^{-4}$ [$1.8 \times 10^{-5}$–$9.4 \times 10^{-4}$] | $4.0 \times 10^{-4}$ [$1.8 \times 10^{-5}$–$9.4 \times 10^{-4}$] | $7.4 \times 10^{-4}$ [$3.4 \times 10^{-5}$–$1.8 \times 10^{-3}$] | $1.0 \times 10^{-3}$ [$4.5 \times 10^{-5}$–$2.4 \times 10^{-3}$] | $2.0 \times 10^{-6}$ [$3.3 \times 10^{-7}$–$7.0 \times 10^{-6}$] |
| **Relative contributions** | | | | | |
| Mean CS | 0.21% | 0.66% | 0.05% | 0.03% | 0.001% |
| Max CS | 0.63% | 2.58% | 0.14% | 0.1% | 0.003% |
| Mean WS | 0.16% | 0.76% | 0.07% | 0.04% | 0.02% |
| Max WS | 0.38% | 1.89% | 0.33% | 0.16% | 0.12% |

Except for As, for which the ambient data were always below the detection limit of 0.25 ng m$^{-3}$, the concentration levels of the toxic elements displayed a strong seasonality, with cold season (Oct. to Mar.) levels roughly twice as high as in the warm season. In the cold season, the observed mean concentrations were 0.14 ng m$^{-3}$ for Cd, 2.21 ng m$^{-3}$ for Ni, and 5.07 ng m$^{-3}$ for Pb; corresponding values in the warm season were 0.07 ng m$^{-3}$, 1.44 ng m$^{-3}$, and 2.93 ng m$^{-3}$, respectively. These concentration levels were slightly lower than those reported in the literature in the vicinity of the Turin WtE plant [28]. The 2 × factor observed between the cold and warm season concentration levels is typical for the area as a consequence of the different atmospheric dispersion conditions. Conversely, the seasonality of BaP data was much stronger, with the cold season average at (0.31 ng m$^{-3}$) about 15 times higher than the warm season (0.02 ng m$^{-3}$), when the concentration levels were very frequently (44%) below the detection limit of 0.005 ng m$^{-3}$. This seasonal pattern is mainly attributable to the extensive use of wooden biomass appliances for residential heating in the area, with a consequent strong contribution to BaP atmospheric presence that originated from its absorption of particulate emissions during colder winter conditions [29,30]. According to the results of a recent survey on the wooden biomass utilization for heating in the Po plain, in the Veneto region, where Schio is located, this type of fuel is used by an average of 30% users over a total population of nearly 2 million inhabitants [31], giving rise to essentially all of the BaP emissions observed for the area, alongside the other sources involved (industrial energy production, road transportation, manufacturing processes) [32]. Indeed, according to emission inventory data for the municipalities within the modelling domain, almost 97% of BaP emissions derive from the use of wooden biomass in the residential heating sector (Table S1).

The estimated contributions of toxic pollutants due to the emissions of the plant were orders of magnitude lower than the measured levels, as summarized in Table 5 and represented in the panels of Figure 6. Seasonally averaged values were 2–3 orders lower for As and Cd, and 3–4 orders lower for Ni, and Pb, without any substantial difference between the cold and warm season. The emissions of the plant had a slightly stronger role on Cd ambient levels than on the other elements, with maximum relative contributions of 2.6% and 1.9% in the cold and warm season, respectively. However, the relative contributions were mostly below 1% (Interquartile range 0.2–1% in the cold season, 0.3–1.1% in the cold season), with absolute concentrations in the 1.8 × 10$^{-5}$–1.6 × 10$^{-3}$ ng m$^{-3}$ range, compared with observed levels in the 5 × 10$^{-2}$–6 × 10$^{-1}$ ng m$^{-3}$ range. On an annual basis, the estimated Cd concentration due to the plant's emissions was in the same order (about 4.6 × 10$^{-4}$ ng m$^{-3}$) reported in a study for a similar plant [19]; however, in our study the relative contribution was higher because of the considerably lower Cd ambient levels (0.1 ng m$^{-3}$ vs. 0.3 ng m$^{-3}$ as annual mean) in the Schio area. Differently from the toxic elements, the estimated relative contribution of the plant to BaP levels was much less (4–5 orders of magnitude), but had a clear seasonal difference due to the abovementioned stronger seasonality of BaP ambient concentrations. Indeed, the plant was responsible for about 2 × 10$^{-6}$ ng m$^{-3}$, representing the cold and warm seasonal average, whereas the ambient concentrations were 0.31 ng m$^{-3}$ and 0.02 ng m$^{-3}$, respectively.

All these results indicate a marginal role of the plant's emissions on the air quality at site R1, without any systematically recognizable impact on the ambient concentration levels for both criteria (i.e., NO$_2$, PM10) and toxic pollutants. Additionally, the strong seasonal variability of BaP ambient levels indicates the relevant impact of the use of biomass for domestic heating, not only as far as airborne PM10 levels are concerned but also with respect to its chemical composition [33]. The small contributions of the WtE plant to the ambient levels resulting from model simulations reflected the information provided by the local emission inventory data for the municipalities within the modelling domain. The annual emissions from the waste sector (i.e., the WtE plant) are in the orders of few percentage points, or of even less than 1% in the case of PM10 and Pb and down to 0.002% for BaP (Table S1). Additionally, some sectors responsible for the larger part of the emissions (i.e., residential heating and vehicular traffic) are characterized by spatially diffused and

ground-level sources, whereas the emissions of the WtE plant derive from an elevated point, thus relying on stronger atmospheric dispersion.

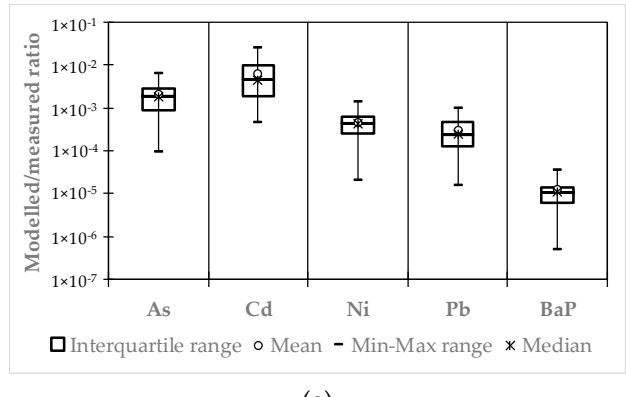 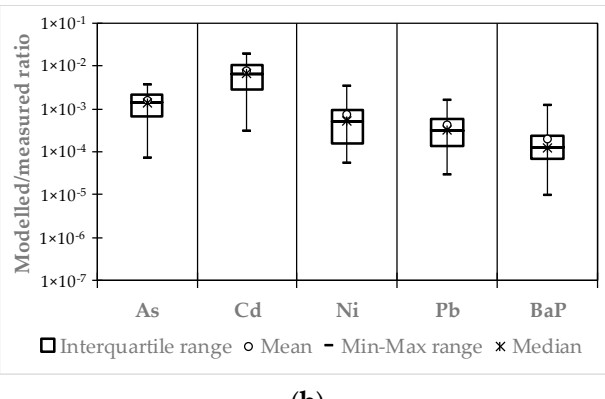

| (**a**) | (**b**) |

**Figure 6.** Distributions of the relative contributions (modelled/measured ratio) of the plant emissions to the daily average concentrations of toxic elements (As, Cd, Ni, Pb) and benzo(a)pyrene (BaP) in the cold (**a**) and warm season (**b**) at site R1.

### 3.2. Monitoring Sites R2, R3, and R4

Concentration data for the four toxic elements and BaP were measured at sites R2, R3 and R4 as weekly averaged values for each individual monitoring campaign. Additionally, the total PCDD/F (i.e., both gaseous and particulate phase) concentration data, in terms of toxic equivalent concentration (TEQ) estimated according to the NATO scheme [34,35], were available with the same time resolution. Conversely, TGM data were available as an hourly time series for the campaigns at sites R2 and R4.

3.2.1. Toxic Elements and Benzo(a)pyrene

The ambient concentration levels measured during the monitoring campaigns were similar at the three sites. Despite the limited dataset extension, this is evidence of a lack of emission sources that are able to affect the air quality at the very local scale, suggesting that the concentration levels are the result of contributions of all the sources distributed over the area. The concentration levels were in good agreement with the seasonal values observed at site R1 and generally displayed the same seasonal variability (Table 6). However, despite being limited to a single week per season, the data from the background site R4 did not show any seasonal difference, except for in terms of BaP concentrations, which were 150 times higher in the cold season than in the warm season (0.64 ng m$^{-3}$ vs. 0.04 ng m$^{-3}$). A strong seasonality of BaP levels was also observed at site R2 (almost 100 times higher in the cold season) and at site R3 (40 times higher). The cold season values (0.40–0.64 ng m$^{-3}$) were all in the order of the maximum concentration at site R1 (0.74 ng m$^{-3}$) and the warm season values (0.004–0.010 ng m$^{-3}$) were similar to the lowest values at site R1, thus leading to such large cold/warm season ratios. Interestingly, as opposed to daily samples at site R1, integrated samples over 1 week allowed us to detect their concentrations, with values ranging between 0.15 ng m$^{-3}$ and 0.93 ng m$^{-3}$.

**Table 6.** Weekly average concentrations measured (Meas.) and modelled (Mod.) during the monitoring campaigns at site R2, R3, and R4 for toxic elements, benzo(a)pyrene and mercury (ng m$^{-3}$), and for dioxins and furans (PCDD/F, fg$_{ITEQ}$ m$^{-3}$).

| Site | Campaign | As | | Cd | | Ni | | Pb | | BaP | | Hg | | PCDD/F | |
|---|---|---|---|---|---|---|---|---|---|---|---|---|---|---|---|
| | | Meas. | Mod. | Meas. | Mod. | Meas. | Mod. | Meas. | Mod. | Meas. | Mod. | Meas. | Mod. | Meas. | Mod. |
| R2 | 4 June–11 June 2018 | <1 | $7.0 \times 10^{-3}$ | <0.2 | $7.0 \times 10^{-3}$ | 1.80 | $1.3 \times 10^{-2}$ | 2.10 | $1.8 \times 10^{-2}$ | <0.02 | $2.8 \times 10^{-5}$ | - | - | 0.2 | $5.0 \times 10^{-2}$ |
| | 11 September–18 September 2018 | 0.32 | $1.3 \times 10^{-2}$ | 0.06 | $1.3 \times 10^{-2}$ | 1.16 | $2.5 \times 10^{-2}$ | 2.20 | $3.4 \times 10^{-2}$ | 0.006 | $5.3 \times 10^{-5}$ | - | - | 6.1 | $9.5 \times 10^{-2}$ |
| | 18 September–26 September 2018 | 0.24 | $1.2 \times 10^{-2}$ | 0.08 | $1.2 \times 10^{-2}$ | 1.52 | $2.2 \times 10^{-2}$ | 1.84 | $3.0 \times 10^{-2}$ | 0.007 | $4.8 \times 10^{-5}$ | 2.3 | $5.6 \times 10^{-3}$ | 3.3 | $8.6 \times 10^{-2}$ |
| | 18 February–25 February 2019 | 0.43 | $5.5 \times 10^{-3}$ | 0.17 | $5.5 \times 10^{-3}$ | 2.21 | $1.0 \times 10^{-2}$ | 5.73 | $1.4 \times 10^{-2}$ | 0.53 | $2.2 \times 10^{-5}$ | - | - | 14.1 | $3.9 \times 10^{-2}$ |
| | 25 February–4 March 2019 | 0.64 | $4.6 \times 10^{-3}$ | 0.25 | $4.6 \times 10^{-3}$ | 4.58 | $8.6 \times 10^{-3}$ | 7.34 | $1.2 \times 10^{-2}$ | 0.51 | $1.8 \times 10^{-5}$ | 1.1 | $2.2 \times 10^{-3}$ | 15.2 | $3.3 \times 10^{-2}$ |
| R3 | 18 September–26 September 2018 | 0.15 | $0.9 \times 10^{-3}$ | 0.05 | $0.9 \times 10^{-3}$ | 2.40 | $1.8 \times 10^{-3}$ | 1.19 | $2.4 \times 10^{-3}$ | 0.010 | $3.8 \times 10^{-6}$ | - | - | 3.6 | $6.8 \times 10^{-3}$ |
| | 25 February–4 March 2019 | 0.61 | $1.7 \times 10^{-3}$ | 0.18 | $1.7 \times 10^{-3}$ | 6.25 | $3.2 \times 10^{-3}$ | 5.79 | $4.3 \times 10^{-3}$ | 0.40 | $6.8 \times 10^{-6}$ | - | - | 11.3 | $1.2 \times 10^{-2}$ |
| R4 | 11 September–18 September 2018 | 0.93 | $0.8 \times 10^{-4}$ | 0.11 | $0.8 \times 10^{-4}$ | 3.23 | $1.6 \times 10^{-4}$ | 4.36 | $2.1 \times 10^{-4}$ | 0.004 | $3.3 \times 10^{-7}$ | 2.0 | $3.9 \times 10^{-5}$ | 2.4 | $6.0 \times 10^{-4}$ |
| | 18 February–25 February 2019 | 0.64 | $1.5 \times 10^{-4}$ | 0.13 | $1.5 \times 10^{-4}$ | 2.60 | $2.8 \times 10^{-4}$ | 4.71 | $3.8 \times 10^{-4}$ | 0.64 | $6.0 \times 10^{-7}$ | 0.9 | $7.0 \times 10^{-5}$ | 17.3 | $1.1 \times 10^{-3}$ |

In contrast to ambient data, the model results were site-dependent according to both the distance of the monitoring site from the plant and the prevailing wind directions during the campaigns (Figure S5). Thus, the highest concentrations were computed for site R2, the closest and mostly downwind the plant, followed by site R3 and site R4. With respect to site R2, the estimated contributions of the plant were from 3 (cold season) to 12 times (warm season) lower at site R3 and from about 30 (cold season) to about 130 times lower at site R4. For the toxic elements, these contributions were in the $0.5–3.4 \times 10^{-2}$ ng m$^{-3}$ range at site R2; $0.9–4.3 \times 10^{-3}$ ng m$^{-3}$ range at site R3; and $0.8–3.8 \times 10^{-4}$ ng m$^{-3}$ at site R4; for BaP, the contributions were, respectively, in the orders of $10^{-5}$, $10^{-6}$, and $10^{-7}$ ng m$^{-3}$. The relative contributions of the plant to the ambient levels were clearly higher at site R2, with average values of 0.2% (Pb), 0.3% (Ni), 1.0% (As), 2.5% (Cd), and 0.004% (BaP) in the cold season, and 1.3% (Pb), 1.4% (Ni), 4.5% (As), 18.3% (Cd), and 0.8% (BaP) in the warm season. Despite the low absolute concentration values, these results confirmed that Cd emissions from the plant were the most concerning, because they produced a relevant contribution to the ambient levels of Cd at the most impacted site R2, especially during the warm season. However, it must be considered that for the Cd simulations, the stack concentration was cautiously assumed to be equal to the detection limit, whereas all measured data were below; therefore, the absolute concentrations and the relative contributions have to be regarded as upper bound estimates. Conversely, for the other toxic elements, and for BaP in particular, the contribution of the plant's emissions was very limited even at site R2.

### 3.2.2. PCDD/F

The concentrations levels of PCDD/F measured during the monitoring campaigns, which were all below 15 fg$_{ITEQ}$ m$^{-3}$, were in agreement with the values registered at the same sites in previous weekly campaigns and with reported data for sub-alpine Northern Italy [36,37]. The concentration levels were well within the 10–50 fg$_{ITEQ}$ m$^{-3}$ range reported for background and unpolluted areas and below the typical range (50–100 fg$_{ITEQ}$ m$^{-3}$) for Europe [38,39]. As for the other pollutants, PCDD/F levels displayed a clear seasonal pattern, such that in the cold season, levels were in the 11.3–14.3 fg$_{ITEQ}$ m$^{-3}$ range whilst warm season levels were in the 0.2–6.1 fg$_{ITEQ}$ m$^{-3}$ range. Overall, cold season concentrations were about four times as high as in the warm season, with the highest 7 × factor at site R4, where the largest cold/warm season ratio was observed for BaP too. The few data available did not allow us to draw robust conclusions for site R4; however, the strong seasonality concurrently observed for BaP and PCDD/F suggested that biomass burning for domestic heating had a meaningful impact at the local scale [40,41]. Indeed, site R4 is located in the centre of a small village at the footsteps of the mountains, where biomass burning is common. Additional evidence of the role of biomass burning was provided by the comparison between the fingerprints (i.e., the relative abundance by mass of the toxic congeners) in the WtE plant's emission and in the ambient air samples (Figure S6). In flue gas samples, in agreement with the literature data [42–45], OCDD (25.7%) was the predominant congener followed by 1,2,3,4,6,7,8-HpCDD (20%); the contributions of all the others congeners ranged between 0.7% (2,3,7,8-TCDD) and 7.9% (2,3,7,8-TCDF). In ambient air samples, OCDD was still the predominant congener (28.3% on the average) followed by 1,2,3,4,6,7,8-HpCDD (12.1%), but with similar contributions from 1,2,3,4,6,7,8-HpCDF (12,2%) and OCDF (14.1%); the contributions of all the others congeners ranged between 0.4% (2,3,7,8-TCDD) and 5.6% (2,3,7,8-TCDF). Indeed, such a relatively high abundance of highly chlorinated furans could be regarded as an indicator of the contribution from biomass burning, because the fingerprint of this source, namely for small domestic appliances, is typically characterized by these latter congeners [46–48].

The estimated contributions of the plant emissions were in the orders of $10^{-4}–10^{-2}$ fg$_{ITEQ}$ m$^{-3}$ and displayed the same features of the other toxic pollutants in terms of spatial and temporal variability. The highest contributions were estimated for site R2 ($3.3–9.5 \times 10^{-2}$ fg$_{ITEQ}$ m$^{-3}$), followed by site R3 ($0.7–1.2 \times 10^{-2}$ fg$_{ITEQ}$ m$^{-3}$), and site R4 ($0.6–1.1 \times 10^{-3}$ fg$_{ITEQ}$ m$^{-3}$). In relative terms, the emissions from the plant were responsible for 0.2–2.6% of the ambient

levels at site R2, for 0.1–0.2% at site R3, and for 0.01–0.03% at site R4; at all the sites, the relative contributions were higher in the warm season due to the lower ambient concentration levels. Concerning the seasonal variability, it is worth noticing that the model output showed a 2 × factor between the cold and warm season results at sites R3 and R4; conversely, rather similar seasonal values at site R2 were observed, with a two-fold higher contribution in the warm season ($8.6$–$9.5 \times 10^{-2}$ fg$_{ITEQ}$ m$^{-3}$ vs. $3.3$–$3.9 \times 10^{-2}$ fg$_{ITEQ}$ m$^{-3}$). This peculiar result, which was observed for toxic pollutants too, might be explained by the different wind conditions during the campaigns. The comparison between the wind roses during the monitoring weeks showed a larger frequency of North-Westerly winds in the cold season weeks (22–26% vs. 14–17% in the warm season, Figure S5). As a result of such wind conditions, in these periods, site R2 was less exposed to the emission of the plant and the estimated concentrations did not show the seasonal variability observed at sites R3 and R4. Conversely, these particular conditions had no effect at the latter sites, because of their location far from the prevailing wind direction, and the observed seasonal pattern of the concentrations, mainly driven by the atmospheric stability, was reproduced.

### 3.2.3. TGM

Total gaseous mercury data were available at sites R2 and R4 as hourly concentrations for one week of the cold and warm season campaigns. As summarized in Table 6, weekly average concentrations measured at the two sites were practically the same (1.1–2.3 ng m$^{-3}$ at site R2, 0.9–2.0 ng m$^{-3}$ at site R4). Concentration data were in line with reported background concentration in the Northern hemisphere (1.5–1.7 ng m$^{-3}$) [49,50]. TGM concentration data displayed the same seasonal variability (2 × factor between warm and cold season data at both sites), but with a statistically significant seasonal variability opposite to the other pollutants. This peculiar feature of TGM ambient levels had already been observed in previous campaigns at the same sites and could be due to a stronger release of mercury from the ground because of the higher average ambient temperature and more intense solar radiation in the warm season [51,52]; however, further investigations are required in order to draw any conclusions. The estimated contributions of the plant emissions were in the order of $10^{-3}$ ng m$^{-3}$ at site R2 and of $10^{-5}$ ng m$^{-3}$ at site R4, reflecting the spatial gradient already observed for the other pollutants. In relative terms, plant's emission had an almost negligible impact on air quality, accounting for 0.20–0.24% of the ambient concentration at site R2 and for 0.002–0.01% at site R4. The difference between measured and modelled concentrations and the extremely low contribution of the emissions of the plant were clearly evidenced by the inspection of the concentration time series on an hourly basis (Figure 7). Despite their being some interruptions in the data series (Site R2), the measured concentrations were fairly constant at both sites, with small fluctuations around the weekly mean values; conversely, the modelled concentrations displayed a great variability, from zero up to 0.1–0.2 ng m$^{-3}$ at site R2 (Figure 7a,b) and to 0.0003–0.0009 ng m$^{-3}$ at site R4 (Figure 7c,d), depending on the meteorological conditions. At site R2, the highest concentrations were associated with the fumigation of the plume of the plant during the nocturnal inversion breakup during morning hours [53,54]; conversely, at site R4, they were mainly determined by the wind condition and associated with south and south-westerly wind that pushed the plume towards this receptor.

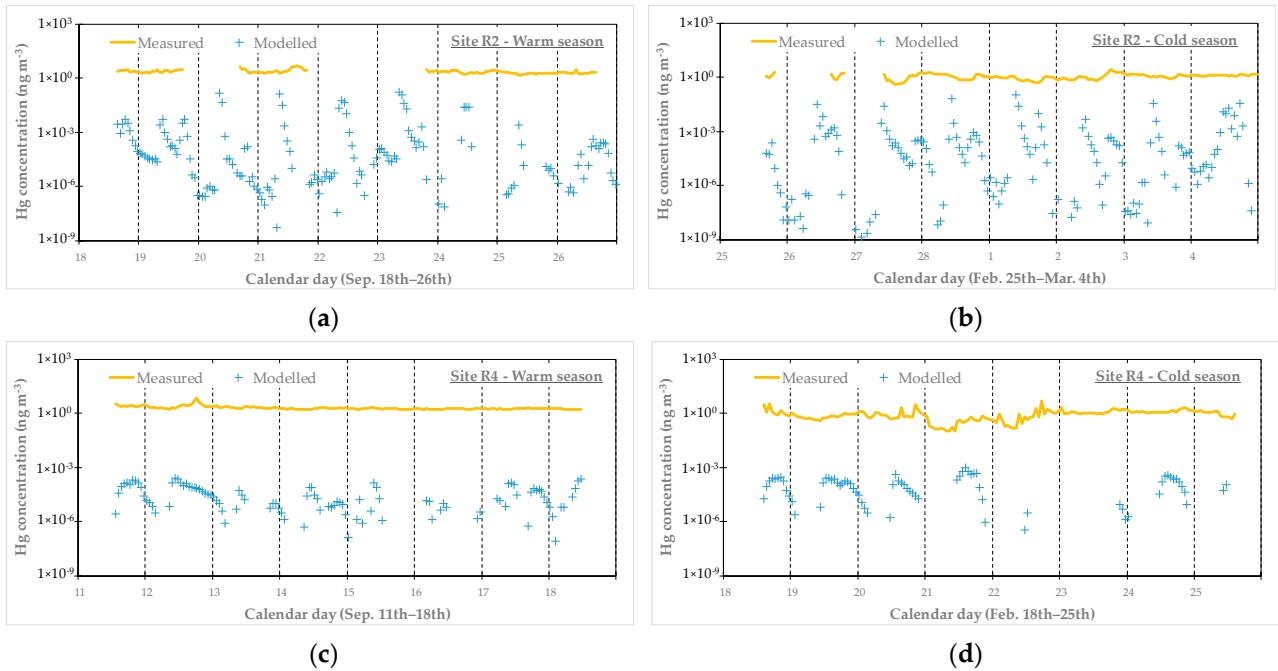

**Figure 7.** Time patterns of measured and modelled 1 h concentration of mercury (ng m$^{-3}$) at site R2 ((**a**): warm season; (**b**): cold season) and R4 ((**c**): warm season; (**d**): cold season).

## 4. Conclusions

The impact on local air quality of the emissions of a municipal Waste-to-Energy plant in Northern Italy was assessed using an atmospheric dispersion modelling with the CALMET-CALPUFF modelling system. Model simulations, based on variable hourly emission rates derived from continuous stack monitoring system data, considered both air quality-regulated pollutants (i.e., nitrogen oxides, particulate matter, toxic elements, benzo(a)pyrene) and other trace pollutants (i.e., dioxins and furans, and mercury). For NOx and PM, the hourly mass flow rates were directly calculated from CEMs data, whilst for the toxic pollutants, which were not continuously monitored, were assessed from the product of the volumetric flow rate from CEMs data and the average concentration measured during the periodic emission monitoring campaigns.

Overall, the contribution of the plant in terms of the maximum mean annual value was more than one order of magnitude (2.7 µg m$^{-3}$ vs. 40 µg m$^{-3}$) lower than the air quality standard for $NO_2$ and up to five orders of magnitude lower for benzo(a)pyrene ($5.8 \times 10^{-5}$ vs. 1 ng m$^{-3}$). For the toxic elements (As, Cd, Ni, Pb), they were at least three orders of magnitude lower than the corresponding limits; for PCDD/F, currently not regulated by air quality standards, the maximum contribution of the plant was three orders of magnitude lower than the 150 fg$_{I-TEQ}$ m$^{-3}$ German guideline value.

The model results were compared with long-term monitoring data from the air quality monitoring regional network and with short-term data from dedicated monitoring campaigns performed at sites in the vicinity of the plant in two periods representative of warm and cold season conditions. Both comparisons showed that the estimated plant contributions are very small. At the network monitoring site, the model results indicate a completely marginal role of the plant's emissions, without any systematic impact on the measured ambient concentration levels for both $NO_2$ and $PM_{10}$, which are the most concerning for compliance with air quality standards, as well as for the regulated toxic pollutants. At the other monitoring sites, the model results displayed some spatial variability, related to both the distance from the plant and the prevailing wind directions during the campaigns, whereas monitored levels were basically similar. This suggests that the observed concentration levels were the result of the contribution of all the sources distributed over the area with residential biomass burning, road transport and some industrial process

activities arising as the most significant contributors according to the emission inventory available for the municipal area of the WtE plant.

The estimated contributions due to the plant's emissions were at least two orders of magnitude lower than the ambient levels at the site closest to the plant and even lower at the sites further away. The role of the plant's emissions was slightly more relevant only for cadmium at the closest monitoring site, with small contributions in absolute terms (always below $1.2 \times 10^{-2}$ ng m$^{-3}$), but accounting for 2.5% (cold season) and for 18.3% (warm season) of the measured concentration because of the extremely low ambient levels (about 10 times smaller than the current air quality standard). However, these contributions were calculated with a conservative value for the stack concentration of cadmium and out of a small dataset of air quality data. Therefore, further concurrent monitoring/modelling results are required to reinforce their significance. Nevertheless, given the very low contributions with respect to the ambient levels, it could be considered that ambient monitoring campaigns might not provide suitable information regarding the real impact of WtE plant's emission in areas with a complex and diversified source activities' regime. Conversely, model simulations, properly developed with real emission data reflecting the actual plant operation conditions across a short time, on an hourly basis, may provide a better representation.

Future research may address some limitations of this work. First, while taking into account the real temporal variability of the emissions, the uncertainty of emission data was not addressed. However, given the accuracy of the monitoring systems and of the analytical methods, we do not expect significant variations in terms of the assessed impact of the WtE plant emissions, even in the worst case scenario in which every single emission value for all the combustion lines was underestimated. Additionally, this work did not address the issue of ground deposition (both wet and dry) of toxic pollutants because deposition measurements were not available. In order to provide comprehensive information to the population living around the WtE plant, future research should also consider this impact pathway. This should include dedicated sampling instruments and longer monitoring periods for atmospheric deposition measurements and, from the modelling side, the size distribution of particulate matter at stack as an input requirement, which is usually not available from stack measurements at plants, with expected large uncertainties in simulation results when it is derived from the few reference literature data available.

**Supplementary Materials:** The following supporting information can be downloaded at: https://www.mdpi.com/article/10.3390/atmos13040516/s1: Figure S1: Distributions of flue gas concentration of Hg, Ni, Pb, BaP, and PCDD/F (all concentrations referred to 0 °C, 101.3 kPa, dry basis, 11% O$_2$). Figure S2: Wind roses at different altitude at the plant location for the simulation reference period (1 May 2018–30 April 2019). Figure S3: Cold and warm season ground-level wind roses at the plant location for the simulation reference period (1 May 2018–30 April 2019). Figure S4: Daytime and nighttime ground-level wind roses at the plant location for the simulation reference period (1 May 2018–30 April 2019). Figure S5: Ground-level wind roses at the plant location for the monitoring campaign periods. Figure S6: Fingerprints (relative abundance by mass) of PCDD/F toxic congeners in the WtE plant's emission and in the ambient air samples. Table S1: Emission inventory data for the municipalities within the modelling domain.

**Author Contributions:** Conceptualization, G.L. and S.C.; Data curation, G.L.; Investigation, G.L.; Methodology, G.L. and S.C.; Writing—original draft, G.L.; Writing—review & editing, S.C. and P.G. All authors have read and agreed to the published version of the manuscript.

**Funding:** This research received no external funding.

**Institutional Review Board Statement:** Not applicable.

**Informed Consent Statement:** Not applicable.

**Data Availability Statement:** The data presented in this study are available on request from the corresponding author.

**Acknowledgments:** The authors wish to thank Maria Sansone of the Environmental Agency of Veneto region (ARPA Veneto, Dipartimento Regionale per la Sicurezza del Territorio—Unità Meteorologia e Climatologia) for running CALMET model simulations and providing input meteorological data for CALPUFF model. Kind acknowledgments are addressed to the support and assistance given by AVA company, responsible of management and operation of the WtE plant, particularly through the cooperation of Riccardo Ferrasin (general direction), Eng. Simone Micheletto (technical management of WtE plant) and Federica Bertoncin (environmental service manager) in providing emission data and air quality data from the dedicated monitoring campaigns.

**Conflicts of Interest:** The authors declare no conflict of interest.

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
