# Peer review of "Air Quality Impact Assessment of a Waste-to-Energy Plant: Modelling Results vs. Monitored Data"

_atmosphere, doi:10.3390/atmos13040516_

Round 1

Reviewer 1 Report

The authors have substantially accepted all the suggested corrections. The cover letter explains in a clear way all the modifications made by the authors and now the paper can be, in my opinion, published.

Author Response

We thank the reviewer for the valuable comments and suggestions.

Reviewer 2 Report

Thank you very much for your work on my manuscript. You have revised the manuscript according to my suggestions. Some minor language mistakes should be revised.

Author Response

We thank the reviewer for the valuable comments and suggestions.

The text was further revised in order to improve its readability and remove some mistakes.

Reviewer 3 Report

This manuscript assesses the impact of a municipal waste-to-energy (WTE) plant on ambient air quality in Schio in Veneto region, Northern Italy, using monitoring data and dispersion modeling. The topic is relevant to the journal of Atmosphere. The revised manuscript addressed most clarification issues raised by the reviewer. However, my major concerns remain the same as in Dec 2021, i.e., there is a lack of data quality control description, in-depth discussion and uncertainty analysis. The presentation quality has room of improvement as well. Overall, the manuscript as written does not denote how this study adds to our body of knowledge on air pollution modeling or air quality management. My specific comments are listed below.

Major concerns

1.There is a lack of data quality control and data screening procedure description. (Dec 2021)

The reviewer could not find any description of data quality control and data screening procedure in the revised manuscript.

  1. The uncertainties of model estimated concentrations should be provided. One source of uncertainty could be the use of average stack parameters and emission rates, noticing the large variability in stack flow rate, exit temperature and velocity (Table 1) and emissions (Table 2). (Dec 2021)

The reviewer could not find any assessment of uncertainty in the revised manuscript.

  1. There is a lack of in-depth discussion. The emissions modeled values are orders of magnitudes smaller than the observed concentrations. The authors may want to discover the reasons, e.g., the emission rates of this plant in comparison with emissions from other nearby facilities. (Dec 2021)

In the revised manuscript, the emissions of the plant were compared with total emissions in the city in a percentage scale. However, the Results section is still rather descriptive, more in-depth discussions of “why” and comparisons with results from other studies might help.

Additional comments (Feb 2022)

Overall, the manuscript as written does not denote how this study adds to our body of knowledge on air pollution modeling or air quality management. Based on the emissions of the plant compared with total emissions in the city in a percentage scale, the concentrations due to emission from the plant are expected low, and be much lower than the long-term monitoring data, and the government standards or guidelines. The simulation presented in this manuscript is often conducted during the permit application stage. It is valuable “to assess the impact of the plant’s emissions on local air quality” and “to make the findings public knowledge for the benefit of the local population” (L64). However, it is not clear whether there is new scientific knowledge obtained from this study. Further, if the monitoring data have been published anywhere in any languages, kindly provide citations. A potential direction is an uncertainty analysis using for example, the Monte Carlo approach.   

Editorial suggestions

There are still quite a few awkward sentences/phrases. The authors may want to enlist the help of a native English speaker to improve the readability.

Author Response

We thank the reviewer for the general and specific comments that helped improving the original manuscript and suggested further developments for future work.

Answer to the general comments

The main goal of our work is the assessment of the real contribution of a WtE plant emissions to the atmospheric levels of organic and inorganic toxic pollutants in its neighborhood, both in the long- and short-term perspective. This goal is achieved by reporting ambient data obtained during dedicated monitoring campaigns and by their comparison with the results of air dispersion modelling for the emissions of the plant.

In general, toxic pollutants data for similar environmental situations are not very common in literature, especially as far as benzo(a)pyrene and gaseous mercury are concerned. Furthermore, the latter was monitored for its vapor phase presence within very short hourly time intervals, thus giving more representative results for its ambient concentration levels with respect to data normally available for the pollutant, generally obtained with measurements on particulate matter over longer sampling periods.

The reviewer commented that “The simulation presented in this manuscript is often conducted during the permit application stage”. Whilst this is, in effect, the most common regulatory requirement for WtE plants, the impacts for such a greenfield situation are evaluated with rather conservative approaches applied in permit applications, involving both flue gas flow rate, evaluated through average and constant values expected at plant maximum operating capacity, and stack concentrations of the pollutants, assumed at their higher values by considering their regulatory emission limits. All these provide maximum impact estimates, coherently with the need to demonstrate the plant’s sustainability in the study area, but do not provide the real contribution of its emissions to ambient concentration levels, that should be derived by simulations based on stack data measurements and environmental levels of the pollutants of interest derived during effective plant operating conditions. Consequently, we cannot agree with the reviewer's opinion in considering our work essentially present in any WtE permit evaluation.

Conversely, we think that our work:

  • provides evidence of atmospheric concentration levels for regulated and non-regulated (dioxins and mercury) toxic pollutants at multiple receptor points nearby a WtE energy plant
  • provides evidence on toxic pollutants concentration in stack flue gas from a WtE plant under real operation conditions
  • provides evidence of the actual contribution of the plant emissions on ambient concentration levels, both with respect to long- and short time periods
  • provides evidence that these contributions are practically negligible with respect to the ambient levels. From this latter point, in areas with complex and diversified source activities’ regime it could be considered that ambient monitoring campaigns might not provide suitable information on the real impact of WtE plant’s emission: model simulations, properly developed with real emission data reflecting the plant operation conditions on short time hourly basis, may give a better representation.

This latter consideration was included in the conclusion section.

Answers to the major concerns

1.There is a lack of data quality control and data screening procedure description. (Dec 2021)

The reviewer could not find any description of data quality control and data screening procedure in the revised manuscript.

We did not operate data quality control and screening because they come from certified monitoring systems. Thus, all available data, both for emission and air quality, were used.

As far as emission data are concerned, the procedures set by EN14181:2014 European standard establishing quality assurance levels for automated CEM systems were adopted by the plant management company to ensure data quality. Data for stack gas from periodic campaigns were produced by a certified laboratory according to ISO and EN standards.

Meteorological data and long-term air quality data were provided by institutional agencies of the Veneto region (ARPAV), where the study area is located. Short-term air quality data were obtained during monitoring campaigns performed according to the current standards for sample collection and analysis.

The text was revised mentioning the quality assurance procedures for CEM data and the adoption of ISO and EN standards for periodic monitoring of the stack emissions.

Reference was added (ARPAV’s report in Italian) for institutional data for short term air quality data.

  1. The uncertainties of model estimated concentrations should be provided. One source of uncertainty could be the use of average stack parameters and emission rates, noticing the large variability in stack flow rate, exit temperature and velocity (Table 1) and emissions (Table 2). (Dec 2021)

The reviewer could not find any assessment of uncertainty in the revised manuscript.

We agree with the reviewer that uncertainty assessment of model estimated concentration is important but this is beyond the scope of our work.

However, concerning emission data, we need to remark that we did not use average stack parameter and emission rates but hourly-variable data (exit gas temperature and velocity, pollutants mass flow rates) derived from continuous monitoring systems for each single combustion line of the plant in order to simulate the real emission regime of the plant during the study period. Data reported in Table 1 are intended to show the variability of the emission features, which are fully considered within the model simulations due to the use of hourly-variable data.

In order to clear this point, we now specify in Section 2 that hourly-variable emission data are used for model simulation of PM2.5 and NOx.

As toxic elements were not measured by the CEM systems, for each single combustion line we used their hourly-variable mass flow rates, obtained by multiplying the hourly variable gas flow rate (recorded by CEMs during our study period) times a constant value for their stack concentration. For a more proper consideration of the variability of stack concentrations, the value utilized is the arithmetic means computed for the lognormal distributions fitted to the concentration values measured during the periodic emission monitoring campaigns.

In our opinion, this approach for the evaluation of emission rates (though not fully “hourly-variable” as for PM10 and NOx) results in the best possible estimation of the real emission rates of toxic elements of the plant, namely considering the actual operation of each combustion line. In the end, we cannot say that “average emissions” were used for the simulation of the toxic elements.

We now specify in Section 2 that hourly-variable emission data were used for model simulation of toxic elements too, providing the explanation on the way these data were calculated.

We think that we were able to properly consider the temporal variability of the emission, relying on monitored data. We might also consider the uncertainty of volumetric flowrates and concentrations, and the subsequent combined uncertainty in mass flow rates. As already stated, this was beyond the scope of this work and could be specifically addressed in other future evaluations.

Anyway, in literature the uncertainties for CEMS flow rate and concentration measurements were reported to be respectively less than 5% and in the 1-1.5% range. Thus, we may expect that the combined uncertainty of emission data is around 6.5%.

According to such rough estimation, even in the worst case scenario in which every single emission value for all the lines was underestimated, the improvements we might have in model results (i.e. the higher concentration values) are not expected to bring significant variations on the assessed impact of the WtE plant emissions.

We revised the conclusion, mentioning the uncertainty analysis as a limitation and future research point, adding the following sentence:

“Future research may address some limitations of this work. First, while taking into account the real temporal variability of the emissions, the uncertainty of emission data was not addressed. Anyway, given the accuracy of the monitoring systems and of the analytical methods, we do not expect significant variations on the assessed impact of the WtE plant emissions, even in the worst case scenario in which every single emission value for all the lines was underestimated.”

  1. There is a lack of in-depth discussion. The emissions modeled values are orders of magnitudes smaller than the observed concentrations. The authors may want to discover the reasons, e.g., the emission rates of this plant in comparison with emissions from other nearby facilities. (Dec 2021)

In the revised manuscript, the emissions of the plant were compared with total emissions in the city in a percentage scale. However, the Results section is still rather descriptive, more in-depth discussions of “why” and comparisons with results from other studies might help.

To our knowledge, works that compare the results of dispersion modelling for WtE plant’s emissions with monitored air quality data are not so frequent in literature. Additionally, modelling approach (i.e. emission rates quantitation, modelling tool) may be different, making sensitive comparison rather difficult. However, we could make reference to the results of a study conducted with the same modeling approach for the emission of a similar WtE plant. The comparison confirms the limited contribution of these plants to ambient concentration levels in absolute terms; however, according to the local features of air quality in the study area, the relative contribution may differ, still remaining very low.  

Comparison with the estimated impacts on air quality for a similar WtE plant was added in the result section.

Answer to the Additional comments (Feb 2022)

Overall, the manuscript as written does not denote how this study adds to our body of knowledge on air pollution modeling or air quality management. Based on the emissions of the plant compared with total emissions in the city in a percentage scale, the concentrations due to emission from the plant are expected low, and be much lower than the long-term monitoring data, and the government standards or guidelines. The simulation presented in this manuscript is often conducted during the permit application stage. It is valuable “to assess the impact of the plant’s emissions on local air quality” and “to make the findings public knowledge for the benefit of the local population” (L64). However, it is not clear whether there is new scientific knowledge obtained from this study. Further, if the monitoring data have been published anywhere in any languages, kindly provide citations. A potential direction is an uncertainty analysis using for example, the Monte Carlo approach.   

We thank the reviewer for the suggestions on Monte Carlo approach for uncertainty analysis. We’ll consider it for future work, even though the combination of hourly variable emission rates with their related uncertainty may give some computational trouble, especially when annual simulations are performed on a 2d domain. In any case, we do not expect that such a refinement of the work could lead to significant variations of the modelled/measured concentration ratios.

Answer to the Editorial suggestions

There are still quite a few awkward sentences/phrases. The authors may want to enlist the help of a native English speaker to improve the readability.

As suggested, we enlisted the help of a colleague from Notre Dame University in the United States in order to have the text further checked and its readability improved.

Round 2

Reviewer 3 Report

This manuscript assesses the impact of a municipal waste-to-energy (WTE) plant on ambient air quality in Schio in Veneto region, Northern Italy, using monitoring data and dispersion modeling. The topic is relevant to the journal of Atmosphere. The revised manuscript added some data quality control descriptions and comparison with the estimated impacts on air quality for a similar WTE plant.  However, the manuscript as written does not denote how this study adds to our body of scientific knowledge on air pollution modeling or air quality management. My specific comments are listed below.

Reviewer comment

Overall, the manuscript as written dose not denote how this study adds to our body of knowledge on air pollution modeling or air quality management. Based on the emissions of the plant compared with total emissions in the city in a percentage scale, the concentrations due to emission from the plant are expected low, and be much lower than the long-term monitoring data, and the government standards or guidelines. The simulation presented in this manuscript is often conducted during the permit application stage. It is valuable “to assess the impact of the plant’s emissions on local air quality” and “to make the findings public knowledge for the benefit of the local population” (L64). However, it is not clear whether there is new scientific knowledge obtained from this study.

Authors’ response

The reviewer commented that “The simulation presented in this manuscript is often conducted during the permit application stage”. Whilst this is, in effect, the most common regulatory requirement for WtE plants, the impacts for such a greenfield situation are evaluated with rather conservative approaches applied in permit applications, involving both flue gas flow rate, evaluated through average and constant values expected at plant maximum operating capacity, and stack concentrations of the pollutants, assumed at their higher values by considering their regulatory emission limits. All these provide maximum impact estimates, coherently with the need to demonstrate the plant’s sustainability in the study area, but do not provide the real contribution of its emissions to ambient concentration levels, that should be derived by simulations based on stack data measurements and environmental levels of the pollutants of interest derived during effective plant operating conditions. Consequently, we cannot agree with the reviewer's opinion in considering our work essentially present in any WtE permit evaluation. concentration levels obtained in the permit application and this work.

Reviewer’s follow up comments (1)

The reviewer agrees with the authors that the operation conditions may differ. However, unless 1) changes in parameters in permit application and during plant operation are identified and quantified, and 2) simulation results in the permit application stage and in this study are compared, the need of carrying out simulation after the permit application does not appear to be justified. This is again because based on the observed emissions of the plant compared with total emissions in the city in a percentage scale, the concentrations due to emission from the plant are expected low, and be much lower than the long-term monitoring data, and the government standards or guidelines.  

Reviewer comment

Further, if the monitoring data have been published anywhere in any languages, kindly provide citations.

Authors’ response

The reviewer could not find any.

Reviewer comment

A potential direction is an uncertainty analysis using for example, the Monte Carlo approach.  

Authors’ response

We thank the reviewer for the suggestions on Monte Carlo approach for uncertainty analysis. We’ll consider it for future work, even though the combination of hourly variable emission rates with their related uncertainty may give some computational trouble, especially when annual simulations are performed on a 2d domain. In any case, we do not expect that such a refinement of the work could lead to significant variations of the modelled/measured concentration ratios.

Reviewer’s follow up comments (2)

An uncertainty analysis may contribute to science instead ofleading to significant variations of the modelled/measured concentration ratios” which could be estimated based on the plant/city emission ratios.  

Reviewer’s additional comments

As mentioned previously, “based on the emissions of the plant compared with total emissions in the city in a percentage scale, the concentrations due to emission from the plant are expected low, and be much lower than the long-term monitoring data, and the government standards or guidelines.” In other words, the use of dispersion model to confirm a conclusion which is easily achievable may not add to our body of knowledge on air pollution modeling or air quality management.  Instead of focusing solely on the contribution of the plant’s emission to local air quality as compared with the corresponding air quality limits/guidelines and ambient concentration levels, the manuscript should test some scientific hypotheses which cannot be answered using the reported emission and concentration data, such as,   

1) What is the seasonal and diurnal (e.g., day/night) variability of the spatial distribution of pollutant concentrations (instead of percent contributions of plant emissions to ambient levels as shown in Fig 4) due to emissions from that plant. This would only require post processing instead of additional simulations.

2) What is the impact of dry and wet deposition on the temporal and spatial distributions of different pollutants? The authors pointed out, “Additionally, this work did not address the issue of ground deposition (both wet and dry) of toxic pollutants because deposition measurements were not available.” (L585). However, incorporating the atmospheric removal (both wet and dry) in the model simulation would not require deposition measurements. It could better reveal the so called “real impact of the plant’s emissions on local air quality” (L65) because depositions of some pollutants are not negligible. Further, the analysis could quantify the difference in the temporal or spatial distribution of concentrations among pollutants due to 1) different emission patterns, and 2) different properties of the pollutant under investigation. 

3) What is the difference in concentration levels obtained in the permit application and in this work? What are the major factors that lead to some large differences, if any, in the concentration levels obtained in the permit application and in this work? See Reviewer’s follow up comments (1) above.

Author Response

Dear Reviewer,

we understand that the main issue you are concerned with is the following:

"the need of carrying out simulation after the permit application does not appear to be justified” because based on the comparison of emission data “the concentrations due to emission from the plant are expected low".

However, as we have already highlighted in the previous revisions, our manuscript is exactly addressing this question. While WtE-related concentrations are expected low from the permit stage, are they actually low when calculated with real emissions data from the stack and compared with ad-hoc measurements? Permit stage modeling neither relies on real emissions data nor measured concentrations, and therefore the rationale (and justification) of our work is to test the hypothesis that WtE concentrations are actually low when using real, measured data.

Additionally, all impact assessment studies do not rely on emission data but on concentration data and assessing the real contribution of the emissions from a source (not as calculated at the permit stage) allows to properly evaluating its impact and associated risks for human health and the environment in general. 

As stated in the introduction, our study “uses a combined modeling-monitoring approach to address the following goals: 

  • to assess the real impact of the plant’s emissions on local air quality based on actual emission data, in order to make the findings public knowledge for the benefit of the local population;  
  • to estimate the contribution of the plant’s emissions to the ambient levels of air quality-regulated pollutants routinely at the regional monitoring network site in the vicinity of the plant; 
  • to compare model results with the ambient levels of non-regulated pollutants (i.e. mercury, dioxin and furans) measured during dedicated air quality monitoring campaigns at sites with different exposure to the plant’s emissions.” 

In order to achieve these goals we relied on (and we make available to the scientific community): 

  • original concentration data for stack emissions (namely for toxic pollutants, reported not only as single point values but as statistical distributions) 
  • original (and unpublished) data for toxic pollutants from seasonal air quality monitoring campaigns around the WtE plant
  • comparison of monitored and modelled data for regulated and non-regulated pollutants. 

We discovered and concluded that: 

  • the real contribution, in terms of ambient concentrations, of the plant’s emission to air quality is small or negligible 
  • monitoring campaign data do not trace the impact of the plant’s emissions, neither at the most exposed receptors, thus raising some questions on the opportunity to perform these campaigns. 

Additionally, you also think that “the manuscript as written does not denote how this study adds to our body of scientific knowledge on air pollution modeling or air quality management” and asks for testing “some scientific hypotheses which cannot be answered using the reported emission and concentration data, such as: 

  1. seasonal and diurnal (e.g., day/night) variability of the spatial distribution of pollutant concentrations 
  2. the impact of dry and wet deposition on the temporal and spatial distributions of different pollutants
  3. the difference in concentration levels obtained in the permit application and in this work”

Remarking that our work was intended to compare model results with measurements, these issues are beyond the scope of our work. 

On point 1), we can say that it is a good suggestion for the analysis of model results, but without any possibility for comparison and validation with measurements.

On point 2), we can say that (as reported in the conclusion as a study limitation, following the comment of reviewer #1) we are not running model sensitivity analysis; moreover, deposition simulation requires input data (i.e. particle size distribution) not available for the our WtE plant, whereas all our work is based on real, monitored emission data without any assumption. 

On point 3), the comparison with permit application results is far away the purpose of the work because we are investigating the real impact of the plant's emissions.

Summing up and concluding, we appreciate your comments and suggestions that we would consider in future works, both from the modelling and the monitoring standpoint. However, at this stage, our work was not performed for improving atmospheric dispersion modelling or introducing new aspects in related simulation science, but in a practical application of an established simulation package for extending, together with dedicated measurement and sampling results, knowledge on the effective impact expected for the plant in its surrounding environment.

This manuscript is a resubmission of an earlier submission. The following is a list of the peer review reports and author responses from that submission.

Round 1

Reviewer 1 Report

This paper describes a modelling work which has the scope to depict the
atmospheric impact of a waste-to-energy plant located in Northern
Italy. The main idea is to compare modelling results with the monitoring data
available around the plant, both in the vicinity and more remotely, in order
to demostrate the small contribution that such plants can have, compared to
the typical pollution levels measured. The simulation, made using the
CALMET-CALPUFF modeling system, was conducted on annual basis, taking into account the emissions of both macro- and micro-pollutants, some of them regulated by air-quality standards. Monitoring data are available both on relatively long-term basis (measured by the stations of the air quality
network managed by the regional environmental authority) and short-term,
messured during some ad-hoc campaign in different seasons.
A positive aspect of the work is related with the use of measured
emission data, leading to a more realistic description of the impact of the
plant.
Generally speaking, the paper is well written and has no particular
shortcomings, even if such kind of simulations, now carried out in large
numbers in impact studies, do not represent a particularly original
scientific contribution. In particular, for some micro-pollutants taken into
consideration it is often more interesting the study of the deposition at
ground (both wet and dry) and it is really a pity that the authors could not take this into account, probably due to the lack of available measurements. In this respect, the work and some of the conclusions appears to be incomplete, also in relation with the idea, that seems to be behind the work, to provide a more correct information to be given to the population, 
that is often worried by the presence of this type
of installations which are considered potentially dangerous for the health. This potential lack should be stated both in the introduction
and in the conclusions of the paper.
Despite this, the paper has a certain usefulness and should be published after some small corrections, which are described in more detail below.

Meteorology
In the description of the meteorological reconstruction, the author should better describe the position of the data that were used inside the computational domain (adding a figure?) to feed
the meteorolological mordel. It seems from the description that the diagnostic reconstruction was made using only
experimental data and that the ones closer to the plant are collected 7 km far from it, a non negligible distance compared to the dimension of the computational domain.
Since the diagnostic system tends to interpolate available data, there is a
risk of reconstructing the flow in the vicinity of the plant in a uncorrect
way. The wind rose illustrated in figure 2, obtained from data extracted in
the vicinity of the plant location, seems in fact to have little to do with the topographical structure around, unless the existence of a breeze cycle could be demosntrated. In this respect I strongly suggest to show in the paper also the wind roses with diurnal-nocturnal and possibly also seasonal separations, informations of this kind are hard to be extracted looking at figure 3. 

Emissions
It should be also useful to add some words related to the emissions of other
possible sources in the region. Is there available any local detailed emission
inventory taking into account the chemical species considered in the
simulation? If yes, a comparison of the orders of magnitude at the higher possible resolution around the plant of the annual emissions of the other sectors with respect to the ones of the plant should be useful. I understand that a simple comparison among the emission flows is not sufficient to determine the difference between the impact at ground (and I also understand that a direct simulation involving the other possible sources is beyond the scope of the paper), but this comparison should help to enforce some stamements in the paper such as:
"This suggests that the observed concentration levels were the result of the contribution  of all the sources distributed over the area and that were not driven by the activity  of one single source" in the conclusion or some others comment related to the contribution of the biomass burning.  

Results
As already written in the introduction, some words about the lack of
both simulated and measured deposition in this study should be added, at least to say that this part should be particularly important to assess the impact of metals, dioxins that could be accumulated in the ground.
Table 4, why only annual averages are taken into consideration for NO2 and
PM10? Could the author add also indexes related to the peak values (such al
the 99.79 percentile of hourly concentrations for NO2 and 90.4 percentile of
daily concentrations for PM10)?

Some minor comments and typos:

Row 27, WtE is written instead of WTE in other points, not fundamental but it should be better to have a uniform
notation

Row 71, "and" instead of "e"

Row 167, "characteristic of the atmospheric turbulence" instead of
"characteristic of the atmospheric stability", to avoud a repetition with
"stability classes" in the same statement

Row 229 "were" instead of "wee"

Row 257, figure caption of figure 5, please use A,B,C,D for uniformity,
uppercases are used in the figures

Table 5. This table has to be somewhat redesigned see the final exponents of
the rows Mod CS and Mod WS

Figure 6, the meaning of the indication "serie1 ....." is not clear

Table 6, similar problem already described for Table 5

Row 446, probably "chares" instead of "share"

Reviewer 2 Report

This is an interesting and well-structured paper. Some suggestions:

  • Introduction: please mention the structure of the paper
  • Introduction: the strong seasonality observed for BaP and PCDD/F suggest that the biomass burning for domestic heating has an important impact at the local scale. The reference of studies related to PCDD/F from forest fires and biomass burning can be done.
  • Some references are not recent
  • Conclusions: Limitations and future lines of research should be mentioned.

Reviewer 3 Report

This manuscript assesses the impact of a municipal waste-to-energy (WTE) plant on ambient air quality in Schio in Veneto region, Northern Italy, using monitoring data and dispersion modeling. The topic is relevant to the journal of Atmosphere. However, there is a lack of data quality control description, in-depth discussion, and uncertainty analysis. The presentation quality has room of improvement as well. My specific comments are listed below.

Major concerns

1.There is a lack of data quality control and data screening procedure description.

  1. There is a lack of in-depth discussion. The modeled values are orders of magnitudes smaller than the observed concentrations. The authors may want to discover the reasons, e.g., the emission rates of this plant in comparison with emissions from other nearby facilities.
  2. The uncertainties of model estimated concentrations should be provided. One source of uncertainty could be the use of average stack parameters and emission rates, noticing the large variability in stack flow rate, exit temperature and velocity (Table 1) and emissions (Table 2).
  3. Figure S2 caption, the reviewer could not find “different altitude”, different time periods are shown instead.

Clarification issues

  1. Please provide the operation schedules of the WTE plant, e.g., 24/7 operation with the same hourly feed rate.
  2. Kindly specify the type of waste being used, e.g., residential, institutional, industrial or a mixture of all the above. Also, if there is any seasonal variability in the feedstock, for example, more organic waste in the warm season, and how the changes in the feedstock would affect the emission rates of some pollutants.
  3. L420, “based on actual hourly emission derived from continuous stack monitoring system data”, kindly clarify whether “hourly emissions from continuous stack monitoring” or “average emission rates derived from continuous stack monitoring” were used in the simulation.
  4. Please specify the emission rates used in the simulation of NO2 and PM2.5, hourly monitored data or average values of monitoring data.

9.Table 2, please clarify the type of data presented, e.g., concentrations.

  1. Kindly define all abbreviations, e.g., “a.s.l.”, “PAHs”

Editorial suggestions

  1. There are quite a few awkward sentences/phrases. The authors may want to enlist the help of a native English speaker to improve the readability. Some examples are listed below,

-“to assess the real maximum impact”

-“grid step”

-“wind (direction and intensity)” and “strengthening of intensity”

-“Time pattern”

-“Model simulations, based on actual hourly emission derived from continuous stack monitoring system data, considered both air quality-regulated pollutants (i.e. nitrogen oxides, particulate matter, toxic elements, benzo(a)pyrene) and other trace pollutants (i.e., dioxins and furans, and mercury).”

  1. International expressions should be used, e.g., “684,28 km, 5064,95 km”, and other places.